# Release and spread of Wingless is required to pattern the proximo-distal axis of *Drosophila* renal tubules

Robin Beaven, Barry Denholm*

Deanery of Biomedical Sciences, University of Edinburgh, Edinburgh, United Kingdom

**Abstract** Wingless/Wnts are signalling molecules, traditionally considered to pattern tissues as long-range morphogens. However, more recently the spread of Wingless was shown to be dispensable in diverse developmental contexts in *Drosophila* and vertebrates. Here we demonstrate that release and spread of Wingless is required to pattern the proximo-distal (P-D) axis of *Drosophila* Malpighian tubules. Wingless signalling, emanating from the midgut, directly activates *odd skipped* expression several cells distant in the proximal tubule. Replacing Wingless with a membrane-tethered version that is unable to diffuse from the Wingless producing cells results in aberrant patterning of the Malpighian tubule P-D axis and development of short, deformed ureters. This work directly demonstrates a patterning role for a released Wingless signal. As well as extending our understanding about the functional modes by which Wnts shape animal development, we anticipate this mechanism to be relevant to patterning epithelial tubes in other organs, such as the vertebrate kidney.

DOI: https://doi.org/10.7554/eLife.35373.001

*For correspondence:
Barry.Denholm@ed.ac.uk

**Competing interests:** The authors declare that no competing interests exist.

## Introduction

The arrangement of epithelia as tubules is a widespread feature of organs. Epithelial tubules transport nutrients, gasses, or fluids, modifying such contents as they transit the lumen. Distinct and ordered functional regions along the tubule proximo-distal (P-D) axis–expressing specific combinations of transmembrane transporters/channels–often underlies this modifying ability. The developmental mechanisms underlying P-D axis patterning in epithelial tubules are not well understood for any organ. The insect renal (or Malpighian) tubule is a relatively simple epithelial tubule that possesses distinct functional regions along its length, including a distal region that secretes the primary urine, a reabsorptive region where the urine is modified, and ultimately a proximal ureter that drains the urine destined for excretion into the gut (references within *Denholm, 2013*; *Sözen et al., 1997*; *O'Donnell and Maddrell, 1995*; *Dow et al., 1994*; *Wessing and Eichelberg, 1978*). We were interested to use this simple system in *Drosophila* to uncover the mechanisms underpinning P-D axis patterning in tubular epithelia.

It is likely that signals emanating from one or other end of a tubule could orchestrate such patterning. Putative roles for several signalling pathways in patterning the tubule P-D axis is hinted at from previous Malpighian tubule studies, or from roles in analogous developmental processes. These include EGF (*Saxena et al., 2014*; *Sudarsan et al., 2002*; *Kerber et al., 1998*), JAK-STAT (*Singh and Hou, 2009*; *Singh et al., 2007*), Notch (*Wan et al., 2000*; *Hoch et al., 1994*), Dpp/BMP and Wnt/Wingless(Wg) (*Lecuit and Cohen, 1997*; *Diaz-Benjumea et al., 1994*).

Wg, which is the focus of this study, plays a role in the establishment of the P-D axis of the *Drosophila* limb and may have been recruited into an analogous role in tubule patterning (*Lecuit and Cohen, 1997*; *Diaz-Benjumea et al., 1994*). A role for the Wnt pathway has also been established in

 

patterning the P-D axis of kidney nephrons (*Lindström et al., 2015*; *Schneider et al., 2015*; *Grinstein et al., 2013*), which may have developmental mechanisms in common with Malpighian tubules. Wg has been shown to be necessary for establishment and proliferation of the Malpighian tubule primordia (*Skaer and Martinez Arias, 1992*), as well as in specifying the tip cell at the distal end of the tubule (*Wan et al., 2000*), however its potential role in patterning the P-D axis of the tubule has not been investigated.

The Wnt/Wg family of developmental signalling molecules have long been considered to act as classical morphogens (*Tabata and Takei, 2004*), spreading through tissues to confer concentration-dependent positional information to cells distant from their source. However the requirement for release and spread of the Wg ligand in developmental patterning has been challenged by more recent studies. Alexandre and colleagues studied flies in which all Wg had been replaced by a membrane-tethered form, which restricts the signal to Wg-producing cells and those cells immediately adjacent. The remarkable normality of the resulting flies, including their normal wing morphology, led the authors to suggest that the release and spread of Wg is not required for patterning, and rather that it appears to function in modulating cell proliferation. As a result they argued that the requirement for the release and spread of Wnts in other contexts should also be revisited (*Alexandre et al., 2014*). It is still a contested question as to whether release and spread is required or is dispensable for Wnt/Wg patterning activity in other tissues.

Here, we show that a Wg signal emanating from the midgut at the proximal end of the Malpighian tubule plays a crucial role in patterning the tubule's P-D axis. In flies in which Wg is replaced by a membrane-tethered form, the proximal tubule is not correctly patterned, leading to severe morphological deformities of the ureter. Wg signalling patterns the proximal tubule by activating expression of *odd skipped* (*odd*), and activation is dependent on an *odd* enhancer, bearing binding sites of the Wg signalling mediator Pangolin/dTCF. This indicates that activation is achieved directly by Wg pathway signalling. Therefore in the Malpighian tubule, in contrast to other fly tissues such as the wing, there is a functional requirement for release and spread of Wg for its patterning activity. To our knowledge this provides the first developmental context in which the release and spread of Wg has been explicitly tested and found to be required for patterning at the level of both gene expression and tissue morphology. This work also uncovers an important mechanism to pattern the P-D axis of an epithelial tube, and establish the Malpighian tubule as an ideal system to study the mechanisms underlying the release and spread of Wnt/Wg in development.

## Results

### Wingless signalling patterns the proximal malpighian tubule

We wished to test the potential roles of different signalling pathways in the P-D patterning of Malpighian tubules during *Drosophila* embryogenesis. To do so we activated or inhibited signalling pathways using a tubule-specific driver (CtB-Gal4 (*Sudarsan et al., 2002*), which is subsequently referred to as *cut*), and analysed the expression pattern of the proximal marker Odd and distal marker Homothorax (*Figure 1B*; *Zohar-Stoopel et al., 2014*; *Tena et al., 2007*; *Ward and Coulter, 2000*). Manipulation of EGF, JAK-STAT, Notch and Dpp/BMP did not produce clear P-D defects (data not shown). However, striking defects in proximal tubule patterning are produced when targeting the Wg pathway.

We firstly tested inhibition of Wg signalling by driving Malpighan tubule expression using a dominant negative version of the pathway regulator *pangolin/dTCF* (*pan/dTCF-ΔN*) (*van de Wetering et al., 1997*). Upon *pan/dTCF-ΔN* expression we observed a greatly reduced Odd-expressing region in the proximal tubule, whilst the midgut expression of Odd (where *pan/dTCF-ΔN* was not driven) was unaffected (*Figure 1B,C*). Secondly we performed the converse experiment in which we activated Wg signalling by driving Malpighian tubule expression of a constitutively active version of the pathway regulator *armadillo/β-catenin* (*arm*$^{S10}$). Upon *arm*$^{S10}$ expression we observed an expansion of the Odd-expressing domain (*Figure 1B,D*).

To rule out the possibility that the expression of *pan/dTCF-ΔN* and *arm*$^{S10}$ affected tubule patterning by disrupting processes other than Wg signalling we assessed Odd staining in a *wg* mutant. As Wg is necessary for tubule cell proliferation at an earlier timepoint we utilised a temperature sensitive *wg* allele (*wg*$^{1-12}$), shifting to a restrictive temperature at the earliest point possible without

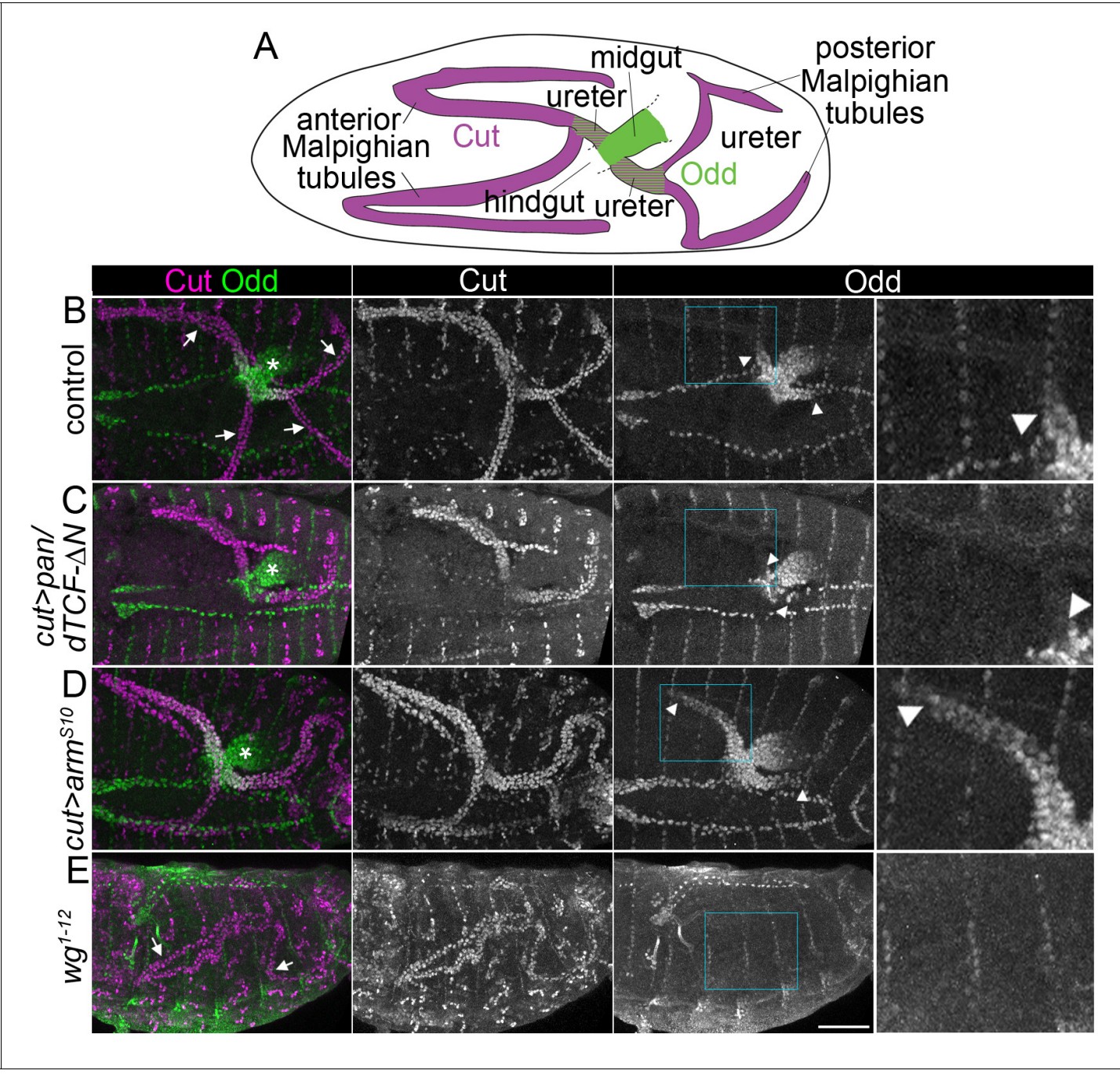

**Figure 1.** Wingless signalling activates expression of Odd skipped in the proximal tubule. (A) Diagram to indicate organisation of the anterior and posterior Malpighian tubule pairs in the embryo, which emanate from the gut at the midgut-hindgut junction. Wild-type expression domains of Cut (magenta) and Odd (green) are indicated. B-E show maximum intensity projections of approximately stage 16 *Drosophila* embryos, oriented anterior left, stained with anti-Cut to mark the entire tubule and anti-Odd to highlight the proximal tubule. Arrowheads mark the distal extent of the Odd-expressing domain in the tubule. Asterisks indicating Odd expression in posterior midgut. An enlargement of the Odd-expression domain in the anterior tubule corresponding to boxed region is provided (right panel). Note that not all tubules may be visible due to orientation. (B) Control (*w[1118]*) embryo showing Odd staining in the nuclei of both ureters at the proximal end of the Malpighian tubules (arrows indicate tubules). (C) Embryo in which dominant negative *pangolin/dTCF-ΔN* (*pan/dTCF-ΔN*) is expressed in tubules under control of the *cut* promoter, to inhibit Wg signalling. (D) Embryo in which constitutively active *armadillo[S10]* (*arm[S10]*) is expressed in tubules under control of the *cut* promoter, to activate Wg signalling. (E) A *wg* temperature sensitive mutant (homozygous *wg[1-12]*), switched to the restrictive temperature during later stages of tubule development (see Materials and methods for timings). Arrows indicate the Malpighian tubules (displaying somewhat abnormal morphology) which show no clear anti-Odd staining. Scale bar = 50 μm.

*Figure 1 continued on next page*

*Figure 1 continued*

DOI: https://doi.org/10.7554/eLife.35373.002

The following figure supplement is available for figure 1:

**Figure supplement 1.** Odd expression in proximal tubule is normal upon loss of zygotic Caudal.

DOI: https://doi.org/10.7554/eLife.35373.003

compromising the final number of tubule cells (*Skaer and Martinez Arias, 1992*; see Materials and methods for details of timing). Under these conditions Odd staining was abolished in the Malpighian tubules and neighbouring midgut (*Figure 1E*). As the domain of Odd expression was lost in *wg* mutants and greatly reduced upon inhibition of Wg signalling, and expanded upon constitutive activation of Wg signalling, we conclude that Odd expression in the proximal tubule is activated by Wg signalling. It is not possible from these experiment to know whether regulation of Odd by Wg signalling is direct or is relayed through other pathways. This issue is addressed with further experiments described below.

We wanted to determine whether Wg functions as a general activator of proximal tubule expressed genes, and therefore assessed the expression of an additional known proximal marker, Caudal/Cdx (*Liu and Jack, 1992*). The expression pattern of Caudal looked normal when we inhibited or activated Wg signalling (*Figure 2A–C*). From this we conclude that an additional pathway, in addition to Wg, is also likely to contribute to the patterning of the proximal tubule.

Our finding that the Odd expression domain was expanded dramatically but did not extend through the entire tubule when Wg signalling (*arm^{s10}*) was activated globally in the tubule (*Figure 1B,D*), suggests this additional signal may also contribute to the activation of Odd. This could be a second signal required to activate Odd in the proximal tubule. As Caudal appears to be functioning in a Wg-independent pathway in the proximal tubule (*Figure 2A–C*) and conserved binding sites of Caudal have been predicted in an enhancer region of *odd* (*Berman et al., 2004*), we considered whether Caudal could function in a second pathway required for activation of *odd* transcription. We therefore examined Odd expression in a *caudal* null mutant, but found that expression looked normal (*Figure 1—figure supplement 1*). We therefore do not find support for the hypothesis that zygotically-expressed Caudal is activating Odd expression in addition to Wg. Our experiment does not rule out the possibility that remaining maternal Caudal could contribute to Odd regulation in the tubule (*Liu and Jack, 1992*).

## A wingless signal emanating from the midgut directly activates odd skipped in the proximal tubule

We sought to determine the source of Wg signalling in the tubule. Firstly, we stained Wg protein using a conventional fixation/staining protocol, which is likely to predominantly reveal the location of intracellular Wg, and to underrepresent the true extent of extracellular Wg (*Strigini and Cohen, 2000*), but we considered it useful for determining the source of the Wg producing cells.

We found Wg expression in the midgut in a region of cells close to the proximal end of the tubule adjacent to the ureter, consistent with previous reports (*Takashima and Murakami, 2001*; *Hoch and Pankratz, 1996*; *Immerglück et al., 1990*; *Baker, 1987*). Wg overlapped with Odd in a small region of the midgut, but the Odd domain extended beyond Wg both further into the midgut and into the tubule ureter (*Figure 3A*). This pattern of expression fits with the idea that Wg activates Odd in the most proximal region of the tubule (and most likely also in the posterior-most region of the midgut, see *Figure 1E*). To attempt to visualise the localisation pattern of extracellular Wg we used a staining protocol previously used for this purpose, in which detergent is excluded (*Piddini et al., 2005*). With this protocol Wg staining appeared qualitatively similar to that described above, but was weaker and more variable (data not shown). We also examined the expression of *frizzled 3 > RFP*, a reporter of Wg pathway activation (*Olson et al., 2011*), by performing staining with anti-RFP. We observed staining throughout the tubule, however this pattern appeared unchanged when we inhibited the Wg pathway by expressing *pan/dTCF-ΔN* using the *cut* driver (data not shown). We consider that RFP may be persisting in tubule cells in which Wg signalling was active at an earlier stage of development, and that *frizzled 3 > RFP* does not give an accurate readout of Wg signalling activity in later development.

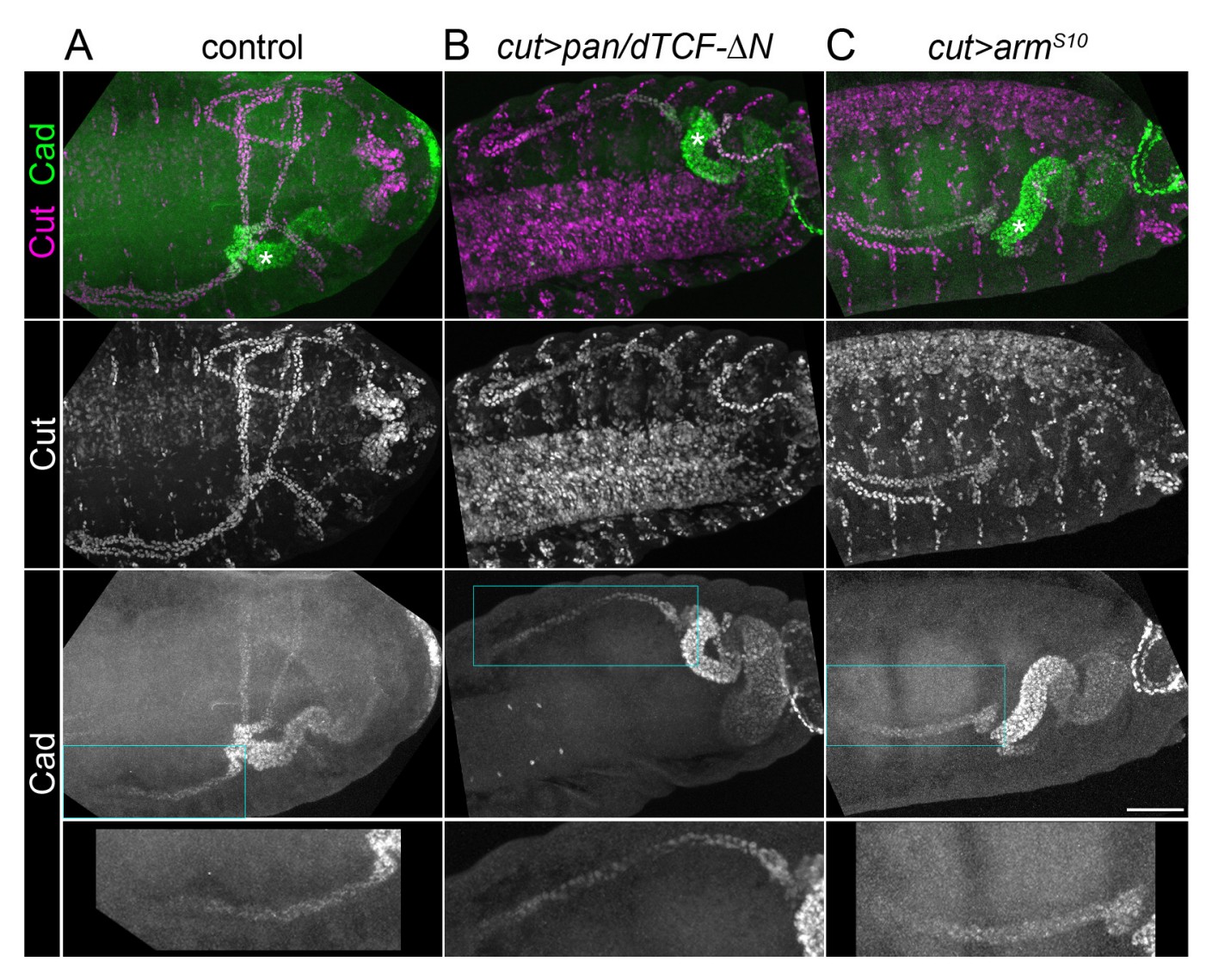

**Figure 2.** Caudal activation in the proximal tubule is independent of Wingless signalling. Maximum intensity projections of approximately stage 16 *Drosophila* embryos, oriented anterior left, stained with anti-Cut to mark the entire tubule and anti-Caudal (Cad). (**A**) Control (*w^1118^*) embryo in which Caudal can be observed staining the nuclei of the posterior midgut and the proximal halves of the tubules Note Cad expression extends more distally than Odd (compare with *Figure 1B*). (**B**) Embryo expressing *pan/dTCF-ΔN* using the *cut* promoter. (**C**) Embryo expressing *arm^S10^* using the *cut* promoter. An enlargement of the Caudal-expressing domain from an anterior tubule corresponding to boxed region is provided (bottom panel). Scale bar = 50 μm.

DOI: https://doi.org/10.7554/eLife.35373.004

We next wanted to determine the mechanism by which Wg activates the expression of Odd, and with this in mind were interested in recent evidence suggesting direct regulation of Odd by Wg. Franz and colleagues identified a 473 bp Wg-responsive enhancer region, 5.2 kb upstream of *odd* (*Franz et al., 2017*). Wg-responsive enhancers typically comprise bipartite recognition sites containing pan/dTCF-binding motifs and distinct Helper motifs (*Archbold et al., 2014*; *Chang et al., 2008*; *van de Wetering et al., 1997*). The 473 bp region contains a Helper motif and three pan/dTCF-binding motifs (*Figure 3B*; *Franz et al., 2017*). In *Drosophila* Kc culture cells it was demonstrated that the ability of this enhancer to drive expression is abolished by mutation of the pan/dTCF motifs suggesting it is a direct target of Wg signalling (*Franz et al., 2017*). We aimed to test whether Wg regulates Odd in vivo. To do so we generated transgenic flies with the 473 bp enhancer region

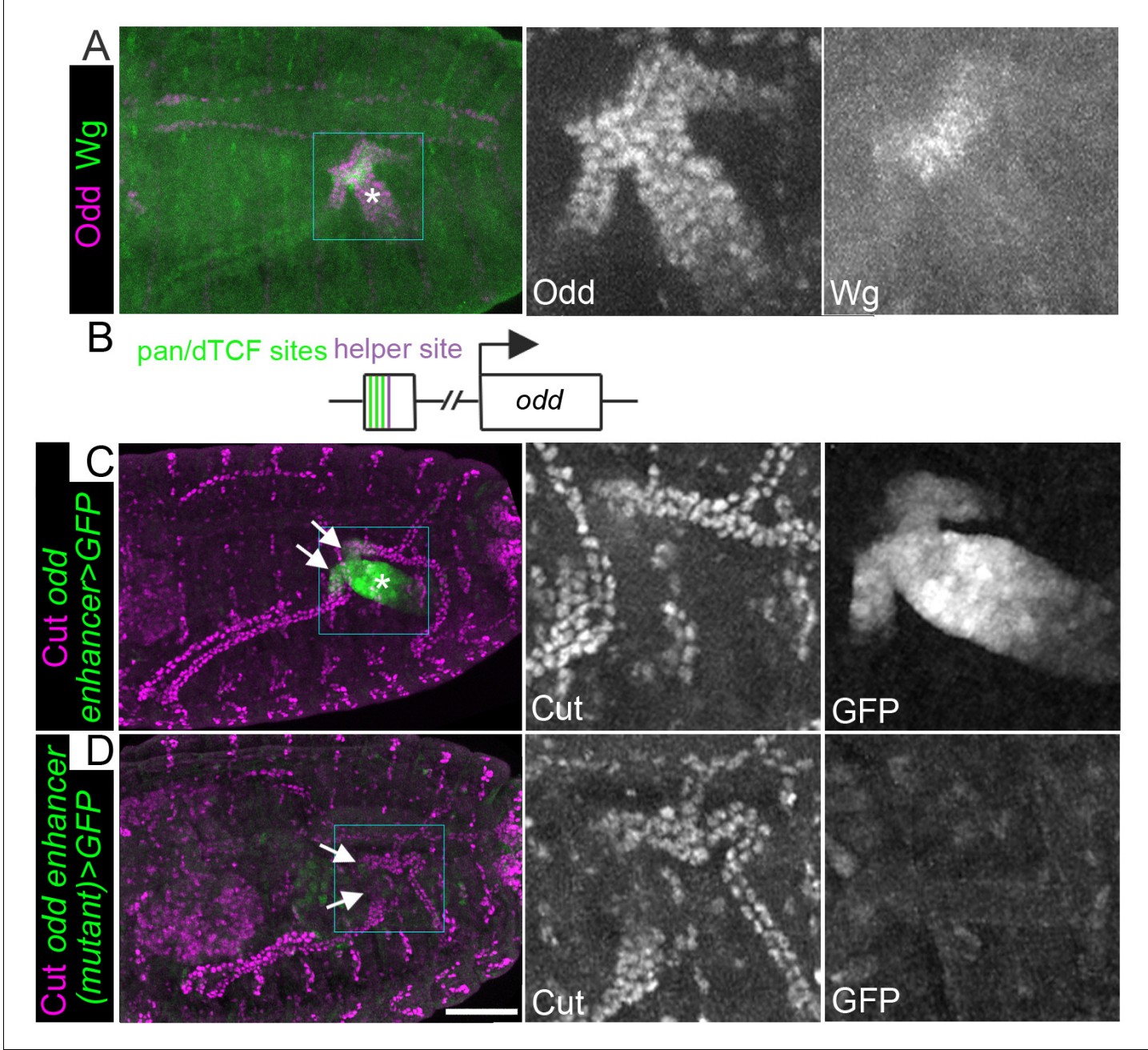

**Figure 3.** Wingless activity extends from the midgut into the proximal tubule to directly activate Odd skipped expression. Maximum intensity projections of approximately stage 16 embryos oriented anterior left. Asterisks indicate midgut and arrows indicate proximal ends of tubules. In each case the two right panels show enlargement of areas marked with squares. (**A**) Staining for anti-Wg and anti-Odd revealing Wg staining in a small patch within the midgut and Odd staining in the posterior midgut and both tubule ureters. (**B**) Cartoon to indicate features of *odd* enhancer region containing pangolin/dTCF binding sites and helper site. (**C**) Transgenic embryo containing the putative *odd* enhancer upstream of a GFP reporter gene (*odd enhancer > GFP*) stained for anti-Cut and anti-GFP. (**D**) Transgenic embryo containing the putative *odd* enhancer bearing mutated pangolin/dTCF binding sites upstream of a GFP reporter gene (*odd enhancer (mutant) > GFP*) stained for anti-Cut and anti-GFP. Scale bar = 50 μm.
DOI: https://doi.org/10.7554/eLife.35373.005

upstream to a GFP reporter. We also generated a version using the previously reported mutant pan/dTCF binding site version (*Franz et al., 2017*). We found that the wild type version recapitulated the Odd expression pattern in the midgut and proximal tubule (*Figure 3A,C*), demonstrating that this enhancer is sufficient to drive expression of *odd* in these tissues. In contrast mutating the pan/dTCF

binding sites abolished reporter gene expression completely (*Figure 3D*). We conclude that Wg signalling acts directly to activate *odd* transcription via binding of pan/dTCF to the *odd* 473 bp enhancer.

## Wingless acts as a released signal to pattern the malpighian tubule

Studies in which Wg is restricted to expressing cells and can only activate signalling in the producing cells and their immediate neighbours (by membrane-tethering of Wg), have demonstrated that Wg is only required as a short-range, autocrine/juxtacrine signal in the context of wing patterning (*Alexandre et al., 2014*). Further, replacing Wg with tethered Wg produced almost entirely normal flies, suggesting that autocrine/juxtacrine Wg signalling is also largely sufficient within the context of the development of the whole animal. However, tethered Wg flies were developmentally delayed and had reduced general fitness when compared to their control siblings. Alexandre and colleagues therefore suggest that although Wg is dispensable for patterning, release and spreading of the Wg signal is required, most likely to modulate cell proliferation.

As we find that *odd* is a direct target of the Wg pathway (*Figure 3C,D*) we can exclude the possibility that intermediate pathways act to relay the Wg signal from its source (a small patch of cells in the midgut) to cells of the proximal tubule (a distance of several cells away). We therefore wondered whether the ability of Wg to be released and to spread from its source would be required for P-D patterning in the tubule. We therefore tested whether tubule expression of Odd was compromised in flies in which *wg* had been replaced by the short-range acting, membrane-tethered version (*Nrt-wg*). In *Nrt-wg* embryos the extent of Odd expression was truncated in comparison to control embryos. The length of the Odd expressing domain measured from the base of the tubule at the midgut was significantly reduced (by ~40%) in the *Nrt-wg* embryos (*Figure 4A–C*). We therefore conclude that there is a requirement for Wg to be released from the Wg-producing midgut cells and to spread over at least several cell diameters into the Malpighian tubule, to correctly pattern the tubule P-D axis.

To ensure that the observed reduction in the Odd domain was not a result of a reduced level of Wg protein in this context we stained *Nrt-wg* embryos with anti-Wg. Although there was variation in Wg staining, it was consistently stronger in batches of *Nrt-wg* embroyos compared to control embryos stained in parallel (*Figure 4—figure supplement 1A,B*), likely to result from an inability of Nrt-Wg to diffuse away from its producing cells.

It is also conceivable that signalling from *wg* expressing cells to their immediate neighbours is in some way disrupted in *Nrt-wg* flies, leading to the reduced Odd expressing domain, although Nrt-Wg has been shown to activate Wg signalling in adjoining cells in the contexts of the wing imaginal disks, and bristle differentiation (*Alexandre et al., 2014*; *Zecca and Struhl, 2010*; *Zecca et al., 1996*). In the midgut of *Nrt-wg* embryos Wg localises at the cell periphery (*Figure 4—figure supplement 1B*), as would be expected for a membrane-tethered construct, indicating that Nrt-Wg is well positioned to signal to neighbouring cells. Furthermore, the domain of cells expressing Odd appears to extend beyond the domain of cells stained with anti-Wg in *Nrt-wg* embryos, suggesting that signalling to neighbouring cells is occurring correctly (*Figure 4—figure supplement 1B*).

A third concern was that the reduction in the Odd-expressing domain could result from dominant effects of Nrt-Wg protein. In order to test this we used *wg-Gal4* to drive expression of *UAS-Nrt-wg* in a wild-type *wg* background. We did not see a reduced length of the domain expressing Odd in this condition, when compared to control embryos or embryos expressing *UAS-wg* (*Figure 4—figure supplement 2B–E*). Surprisingly we observed a slightly increased Odd-expressing domain when expressing *UAS-Nrt-wg* as well as more distal patches of cells expressing Odd (and also in the hindgut). This is likely to be explained by the finding that *wg-Gal4* appears to be active along the tubule (and hindgut) as well as in the proximal midgut (*Figure 4—figure supplement 2A*). This may result from perdurance of Gal4 as the tubule cells are known to express *wg* earlier in tubule development. The key conclusion from this experiment is that the presence of Nrt-Wg does not seem to have a dominant effect in disrupting Odd expression.

To determine the functional consequences of Wg patterning we assessed tubule morphology in tethered Wg versus control adults. In wild-type animals the two sets of tubules (anterior and posterior) join at their proximal-most end in a common ureter, which inserts into the gut at the midgut-hindgut boundary, tapering slightly toward the point of insertion. The ureter is ensheathed in visceral muscle (*Figure 5A*; *Sözen et al., 1997*; *Skaer, 1993*; *Campos-Ortega and Hartenstein, 1985*;

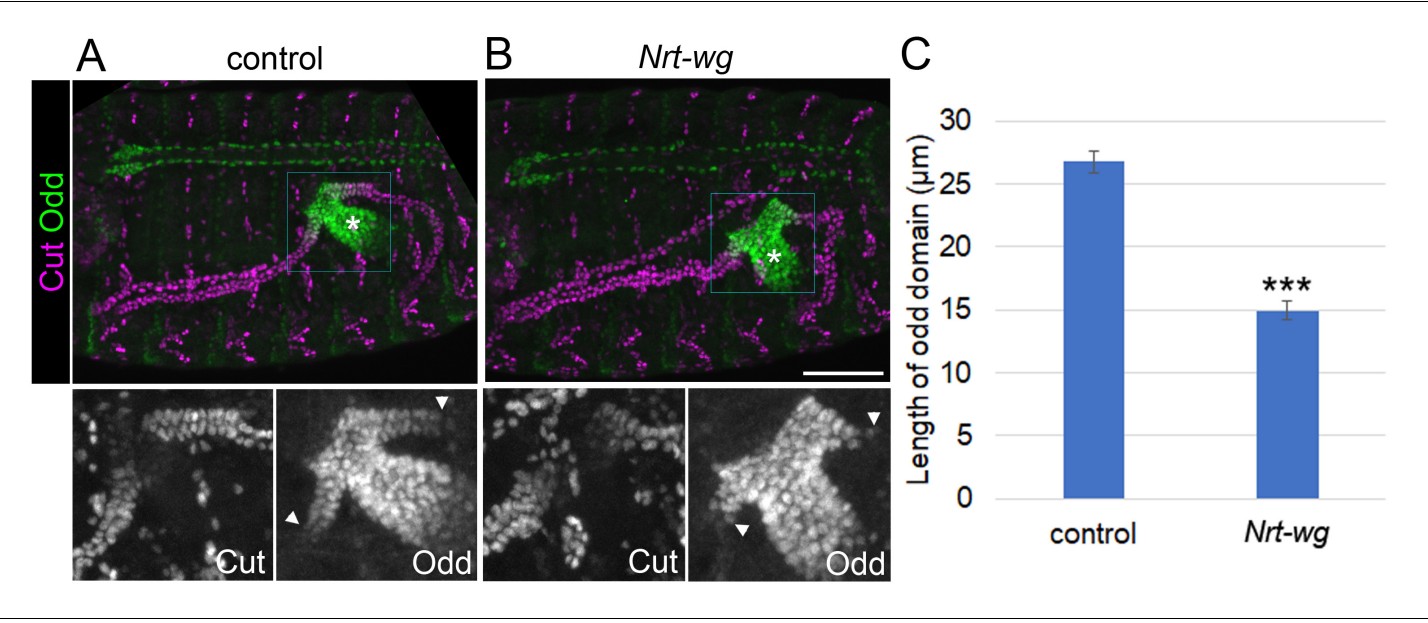

**Figure 4.** Membrane-tethered Wingless is unable to correctly pattern the proximal tubule. Maximum intensity projections of approximately stage 16 embryos stained with anti-Cut and anti-Odd, oriented anterior left. Asterisks indicate midgut and arrowheads mark distal extent of Odd expression in the tubule. Lower panels show enlargement of boxed areas. (A) Sibling control embryo (*Nrt-wg/+* or *+/+*). (B) Embryo which is homozygous mutant for *wg*, and expressing membrane-tethered *Nrt-wg*. Scale bar = 50 μm. (C) Graph showing mean length of Odd expressing region along tubule in control (mixture of *Nrt-wg/+* and *+/+*) and *Nrt-wg* embryos. Error bars indicate SEM. Control n = 29, *Nrt-wg* n = 37. P (Mann-Whitney) <0.00001***.

DOI: https://doi.org/10.7554/eLife.35373.006

The following source data and figure supplements are available for figure 4:

**Source data 1.** Quantifications of Odd skipped domain length in *Nrt-wg* embryos.
DOI: https://doi.org/10.7554/eLife.35373.011
**Figure supplement 1.** Expression of *Nrt-wg*.
DOI: https://doi.org/10.7554/eLife.35373.007
**Figure supplement 2—source data 1.** Quantifications of Odd skipped domain length upon overexpression of *Wg* or *Nrt-Wg*.
DOI: https://doi.org/10.7554/eLife.35373.009
**Figure supplement 2.** Incorrect patterning of proximal tubule is not result of dominant effect of *Nrt-wg*.
DOI: https://doi.org/10.7554/eLife.35373.008
**Figure supplement 3.** Apoptosis is not induced in tubules of *Nrt-wg* embryos.
DOI: https://doi.org/10.7554/eLife.35373.010

*Wessing and Eichelberg, 1978*). We observed that ureter morphology was strikingly abnormal in *Nrt-wg* compared to controls. *Nrt-wg* ureters were significantly shorter than control ureters (~200 um, n = 30 versus~375 um, n = 47 respectively) (*Figure 5A–C*), and tapered much more dramatically, having a bloated distal section which was often devoid of surrounding visceral muscle (*Figure 5A,B*). We do not consider it likely that an elevated level of cell death contributes to the *Nrt-wg* phenotype as there was no evidence of apoptosis in the proximal tubule of *Nrt-wg* embryos (*Figure 4—figure supplement 3*).

This phenotype is likely to involve an interplay between the epithelial cells of the tubule, and the visceral muscle cells which surround them. It seems likely that the identity of the ureter cells is altered in *Nrt-wg* adults, for example through an inability to correctly activate the expression of Odd, and/or potentially other targets of Wg signalling. This phenotype could then compromise their recognition by visceral muscle cells as they migrate onto the ureter during development, leaving them devoid of ensheathing visceral muscle. This could result in an inability of the distal ureter to extend to its usual long, thin morphology due to its altered mechanical environment. In support of this model, ablation of the visceral muscle has been shown to disrupt normal extension of the midgut during pupal metamorphosis (*Aghajanian et al., 2016*). Additionally we find that constitutive activation of Wg signalling by driving the expression of *arm*[S10] in the tubules gives rise to adults in

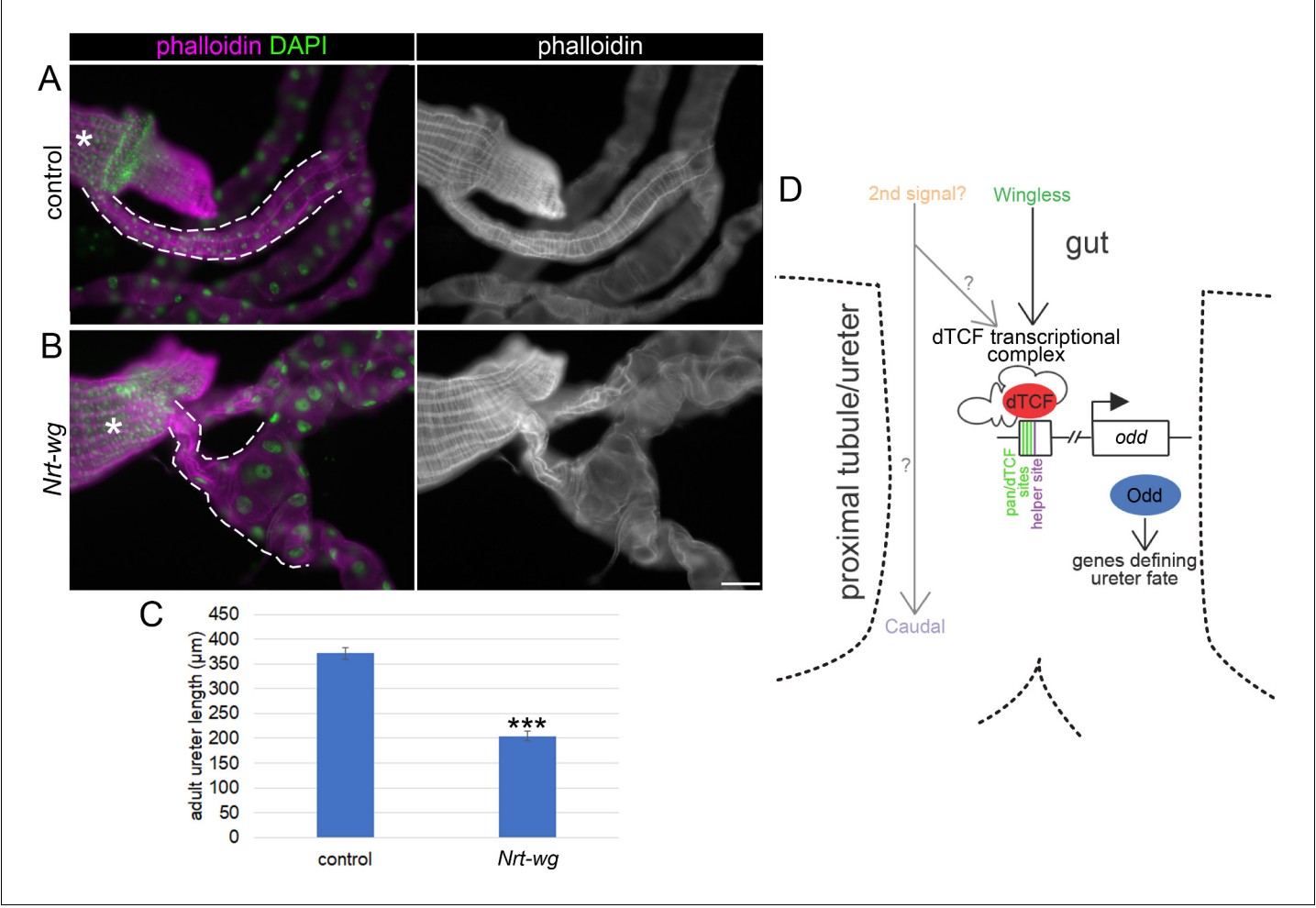

**Figure 5.** Disruption of the Wingless pathway leads to abnormal morphology of the adult ureters. Single focal plane epifluorescence images of dissected guts and Malpighian tubules of adult flies, stained to show actin (phalloidin) and nuclei (DAPI). White dashed line indicates most proximal tubule/ureter and asterisks indicate midgut. (**A**) Centred on ureter from sibling control (*Nrt-wg*/+) adult. (**B**) Centred on ureter from fly which is homozygous mutant for *wg*, and expressing membrane-tethered *Nrt-wg*. Scale bar = 50 μm. (**C**) Graph showing mean length of adult ureters in control (*Nrt-wg*/+) and *Nrt-wg* flies. Error bars indicate SEM. Control n = 47, *Nrt-wg* n = 30. P (Mann-Whitney) <0.00001***. (**D**) Model of signals from the gut patterning the proximal Malpighian tubule, necessary for correct ureter development. Enhancer region upstream of *odd* coding region as in *Figure 3B*.
DOI: https://doi.org/10.7554/eLife.35373.012

The following source data and figure supplement are available for figure 5:

**Source data 1.** Quantifications of adult tubule ureter length in *Nrt-wg* flies.
DOI: https://doi.org/10.7554/eLife.35373.014

**Figure supplement 1.** Changes to morphology of adult tubules is not the result of a dominant effect of *Nrt-wg*.
DOI: https://doi.org/10.7554/eLife.35373.013

which the visceral muscle extends further distally over the Malpighian tubule surface (*Figure 5—figure supplement 1*), providing further support for the notion that Wg signalling can alter the identity of tubule cells, which in turn regulates their interactions with visceral muscle cells.

Together our data indicate that a released Wg signal is required to pattern the P-D axis of the tubules to establish normal ureteric morphology.

## Discussion

Since the 1970s, morphogens have been conceptualised in light of Wolpert's positional information/French Flag model, and similar concepts proposed by Lawrence and by Crick. In this view

morphogens are signalling molecules, released from their producing cells to form a graded distribution across a tissue, resulting in different domains of gene expression depending on the activation thresholds of these target genes (references within *Green and Sharpe, 2015*; *Lawrence et al., 1972*; *Crick, 1970*; *Wolpert, 1969*). Wnt/Wg appeared to fit the bill as a classical morphogen, and has long been considered as such (*Tabata and Takei, 2004*; *Neumann and Cohen, 1997*; *Zecca et al., 1996*; *Hoppler and Bienz, 1995*; *Struhl and Basler, 1993*). However, the surprising finding that membrane-tethered Wg is able to substitute for wild type Wg to produce a largely normal fly suggests Wg primarily functions as an autocrine/juxtacrine signal and need not extend beyond the cells that produce it and their direct neighbours (*Alexandre et al., 2014*; *Pfeiffer et al., 2000*). A number of earlier studies had also provided evidence in this direction (*Couso et al., 1994*; *Diaz-Benjumea et al., 1994*; *Vincent and Lawrence, 1994*; *van den Heuvel et al., 1989*; *Martinez Arias et al., 1988*).

In some cases where Wnt/Wg acts at a distance from its source, it has been shown that this occurs via cell movement rather than by release and spreading of the Wnt/Wg ligand, and is therefore functioning as an autocrine/juxtacrine signal rather than a morphogen as classically defined. For example in the case of segmental patterning in *Drosophila* embryos, Wg is transcribed in a single cell stripe, but forms a protein gradient that spans several cells. However using *Nrt-wg* embryos it was shown that this gradient is generated by cell division and movement, rather than by release and spreading of Wg (*Pfeiffer et al., 2000*). Similar mechanisms have since been identified in vertebrates, for example in the mammalian intestinal stem cell niche, where Wnt3 predominantly generates a gradient through cell division and movement (*Farin et al., 2016*), or in the presomitic mesoderm where a Wnt gradient is established in a similar manner in order to specify somite segmentation (*Aulehla et al., 2003*). Additionally, long-range Wnt signalling to somites occurs by migration of neural crest cells (*Serralbo and Marcelle, 2014*).

The conflicting views of Wnt/Wg as a morphogen or an autocrine/juxtacrine signal demonstrate how remarkably open the question of the mechanism by which Wnt/Wg pattern tissues remains. The work described here adds to this debate providing direct functional evidence that release and spread of Wg is required for the P-D patterning of the Malpighian tubule. Crucially, as we found that the membrane tethered form of Wg is not sufficient to correctly pattern the tubule, we demonstrate that cell division and movement cannot explain the propagation of the Wg signal along the tubule. Wg exerts its influence on a broader domain of the tubule than those cells in immediate contact with the Wg producing cells. The distal extent of Odd expression in the tubule most likely determined by a threshold in a Wg gradient declining with distance from the midgut. It remains to be seen whether other Wg target genes exist in the tubule, with different activation thresholds. If so Wg would serve as a true classical morphogen in this context.

We note that in other respects the Malpighian tubules are remarkably normal in the *Nrt-wg* condition. So the roles Wg plays during earlier steps of tubule development, for example in generating the tip cell and supporting cell division (*Wan et al., 2000*; *Skaer and Martinez Arias, 1992*) seem to only require autocrine/juxtacrine Wg signalling. It therefore appears that Wg, and most likely other Wnts, function as released signals in some developmental contexts and as autocrine/juxtacrine signals in others, and even in the development of a single organ system might utilise these two functional modes at different developmental stages.

It will be important to determine whether Wnt/Wg functions as a released verses an autocrine/juxtacrine signal in each individual development/physiological context. Patterning of the animal A-P axis is one area where previous studies suggest a likely role for Wnt as a classical morphogen (*Hikasa and Sokol, 2013*). As Wg does not play a major role in this context in *Drosophila* (*Vorwald-Denholtz and De Robertis, 2011*), this could explain the relative normality of membrane-tethered Wg flies. Another context where we anticipate Wnt to act as a released signal is in patterning the nephrons of the kidney, which are also epithelial tubules. Canonical Wnt signalling, and the Wnt pathway component β-catenin, have been implicated in patterning the P-D axis of the vertebrate nephron (*Lindström et al., 2015*; *Schneider et al., 2015*; *Grinstein et al., 2013*). Similar mechanisms could also be important in other epithelial tubules in which Wnt signalling has been implicated, for example in developing lungs (*McCauley et al., 2017*; *Shu et al., 2005*), and salivary glands (*Patel et al., 2011*).

In nephron patterning it has been noted that a gradient of Wnt signalling activity is present along the P-D axis, although it is a morphologically convoluted structure. A mechanism that has been

proposed to explain this, is that a gradient of released Wnt may be restricted within the lumen of the nephron, rather than spreading 'as the crow flies' from the Wnt source (*Lindström et al., 2015*). We suggest a similar mechanism corrals Wg-dependent activity to the proximal tubule; this would be particularly important at later stages of development when the tubule forms a convoluted hairpin structure with the distal-most end of the anterior tubule is in close proximity with the Wg source in the midgut (e.g. *Figure 1*). Such shaping of a signal's distribution by the tissue architecture also echoes recent reports, for example during development of the intestinal epithelium in chicks and mice tissue folding shapes the distribution of Shh, to define which cells ultimately become stem cells (*Shyer et al., 2015*), and the trapping of FGF in luminal structures establish the position where mechanosensory organs develop in zebrafish (*Durdu et al., 2014*).

There is emerging evidence of mechanisms by which Wnt/Wg can be targeted to its receiving cells, for example by cytonemes and other filopodial like projections (*Huang and Kornberg, 2015*; *Sagar et al., 2015*; *Stanganello et al., 2015*; *Luz et al., 2014*), exosomes and lipoprotein particles (*Beckett et al., 2013*; *Gross et al., 2012*; *Korkut et al., 2009*; *Neumann et al., 2009*; *Panáková et al., 2005*; *Greco et al., 2001*), or through interaction with secreted chaperone proteins (*Mulligan et al., 2012*; *Mii and Taira, 2009*). Here we establish the Malpighian tubule as an ideal platform for future mechanistic studies of how spreading of the released Wg signal occurs, which we anticipate to be relevant for understanding this mode of Wnt/Wg function in other developmental contexts.

Vertebrate Osr-1, an orthologue of Odd, is also known to function in nephron development, although putative roles in P-D axis patterning have not been established (*Xu et al., 2014*; *Tena et al., 2007*; *James et al., 2006*; *Wang et al., 2005*). Existing evidence shows that Osr-1 blocks Wnt driven nephron differentiation to maintain a pool of nephron progenitor cells, by inhibiting transcription of Wnt's downstream targets (*Xu et al., 2014*). This differs from the regulatory mechanism we have uncovered in Malpighian tubules. However the molecular networks of Wnt/Wg pathways and Osr/Odd members are likely to be complex, for example differing expression patterns and functions have been demonstrated for different Wnt members in the developing kidney (*Levy-Strumpf, 2016*; *Miller and McCrea, 2010*; *Green et al., 2008*; *Merkel et al., 2007*; *Weidinger and Moon, 2003*; *Itäranta et al., 2002*; *Lin et al., 2001*). Furthermore, *Osr* members can be found in microarray data from mouse kidneys in which Wnt signalling has been blocked, with *Osr-1* and *Osr-2* being upregulated and downregulated respectively (*Bridgewater et al., 2008*). It therefore seems possible that mammalian Osr-2 functions in a role analogous to *Drosophila* Odd, in patterning an epithelial tubule as a direct target of Wnt signalling.

We found that Wg signalling from the midgut directly regulates Odd expression in the proximal tubule, through the binding of pan/dTCF to an upstream enhancer region of *odd* (*Figure 5D*). It is also likely that the Wg pathway acts alongside a further pathway(s) to pattern the proximal tubule. Firstly, it seems that a proximal transcription factor, Caudal, functions in an independent pathway to the Wg pathway (although we have not ruled out the possibility that maternal Caudal acts upstream of Wg), as expression of Caudal was not altered upon Wg signalling perturbation (*Figure 2A–C*). Secondly, we found that activating Wg signalling in all tubule cells extends the Odd expression domain, but not throughout the entire tubule (*Figure 1B,D*), which could suggest that an additional pathway, along with Wg, is required to activate Odd expression in the proximal tubule. The pathway that actives Caudal may also be required to activate Odd. As the 473 bp *odd* enhancer is sufficient to recapitulate the expression pattern of *odd* in the midgut and proximal tubule, the simplest model is that the additional pathway also acts within this enhancer region to upregulate *odd* expression. This pathway could act by regulating a component of the pan/dTCF containing transcriptional complex (e.g. a co-activator), enabling it to drive *odd* expression (*Figure 5D*). Examples of such crosstalk have been reported before, for example β-catenin, a component of the pan/dTCF transcriptional complex, can be activated downstream of EGF signalling (*Ji et al., 2009*), and Smad proteins in the TGFβ cascade can assemble with the pan/dTCF transcriptional complex to regulate expression of specific target genes (*Labbé et al., 2000*; *Nishita et al., 2000*). How pathways might crosstalk with the Wg pathway to pattern the proximal tubule is a fascinating question for future studies.

## Materials and methods

### Fly stocks

Flies were cultured on standard media at 25°C unless otherwise stated. Control flies were from a *w*$^{1118}$ stock, or in cases where mutants were analysed, mutations were balanced over a *YFP* bearing chromosome (*dfd-GMR-nvYFP* balancers [*Le et al., 2006*]), and *YFP* expressing embryos were used as control. The following stocks were also used: a mixture of embryos bearing one or two copies of a *cut* driver, to drive Malpighian tubule expression (*CtB-Gal4* [*Sudarsan et al., 2002*]), *UAS-pangolin/dTCF-ΔN* (*van de Wetering et al., 1997*), *UAS-armadillo*$^{S10}$ (*Pai et al., 1997*), a temperature sensitive allele of *wg* (*wg*$^{1-12}$) for which egglays were carried out for 2 hr at a permissive temperature (18°C), before incubation for ~14 hr at 18°C and finally incubated for ~7.5 hr at the restrictive temperature (25°C) which should result in depletion of Wg at later stages without inhibiting tubule outgrowth/proliferation (*Skaer and Martinez Arias, 1992*), an amorphic *caudal* mutant (*cad*$^2$ [*Jürgens and Weigel, 1988*]), a section of the *frizzled three* promoter driving *RFP* expression (*fz3 > RFP* [*Olson et al., 2011*]), *Nrt-wg*, a *wg* mutant bearing a transgenic insertion of *wg* fused to *Nrt* (*wg{KO; NRT–Wg; pax-Cherry}* [*Alexandre et al., 2014*]), *UAS-Nrt-wg* (from Jean-Paul Vincent), *P{UAS-wg.H.T:HA1}6C* (Bloomington stock 5918), *wg-Gal4* (referred to as wg{KO; Gal4} [*Alexandre et al., 2014*]), *UAS-CD8-GFP* was used as an expression reporter. For generation of *odd enhancer > GFP* flies, the wild type or mutant plasmid (see Molecular cloning section below) was integrated onto the third chromosome by phi mediated recombination, by injection into *y[1] w [1118]; PBac{y[+]-attP-3B}VK00033* flies, performed by BestGene Inc.

### Embryo fixation and staining

Embryos were collected on grape juice agar plates with yeast paste. Embryos were fixed and antibody stained using standard techniques (*Weavers and Skaer, 2013*) with the following antibodies: anti-Cut (mouse, 1:200, 2B10-c from DSHB), anti-Odd (rabbit, 1:400, from J. Skeath), anti-Caudal (rabbit, 1:200, from Paul MacDonald), anti-Wg (mouse, 1:200, 4D4, from DSHB), anti-RFP (rabbit, 1:500, ab62341, from Abcam), anti-GFP (to visualise presence of YFP carrying balancers, goat, 1:500, ab6673, from Abcam), anti-cleaved Dcp-1 (rabbit, 1:100, Asp216, Cell Signaling technology). Secondary antibodies from Jackson ImmunoResearch of the appropriate species tagged with 488, Cy3 or Cy5 fluorophores were used. Embryos were mounted in Vectashield (Vector Laboratories) or 85% glycerol, 2.5% propyl gallate, and imaged using either a Nikon A1R or Zeiss LSM800 confocal microscope. Maximum intensity projection images were generated using Fiji.

### Adult tubule fixation and staining

Adult guts with attached tubules were dissected in PBS and transferred to 4% PFA in PBS on ice during the collection, fixed on a shaker at room temperature for a further 20mins and rinsed with PBS. Washes were performed in PBS + 0.3% Triton X-100 + 0.5% BSA. Stainings were performed with phalloidin-568 (1:100, A12380, Molecular Probes) and DAPI (1:1000, Molecular Probes) for 1 hr on a shaker, before finally washing and mounting in 85% glycerol, 2.5% propyl gallate. Imaging was performed on a Leica DMR epi-fluorescence microscope (*Figure 5A,B*). For *Figure 5—figure supplement 1A–C* a Zeiss LSM800 confocal microscope was used and maximum intensity projection images were generated using Fiji.

### Molecular cloning

To generate plasmids for *odd enhancer > GFP* (wild type and pangolin/dTCF binding site mutant) flies, pGreenRabbit (a reporter vector carrying GFP under a minimal promoter, and flanking insulator elements to reduce influence of neighbouring sequences, *Housden et al., 2012*) was digested with NotI (New England Biolabs) and gel purified. Plasmids carrying an upstream enhancer region of *odd* (wild type and pangolin/dTCF binding site mutant [*Franz et al., 2017*]) were used as PCR templates, and inserts were amplified using TCTAGAGGATCCCCGCACCAGCGAACCACTGAACCAC (forward) and ATCCCCCGGTACCCGCATCGATCGGGCGGCAGTCACCAT (reverse) primers. Each PCR product was then joined to the linearised pGreenRabbit plasmid using Gibson assembly (New England Biolabs).

## Acknowledgements

We are thankful to the following people and organisations for their help: Jamie Davies, Helen Skaer and Peter Hohenstein for critical reading, Jean-Paul Vincent, Golnar Kolahgar and the Bloomington Drosophila Stock Center and Resource Center (National Institutes of Health grants P40OD018537 and 2P40OD010949-10A1) for fly stocks, Paul MacDonald for antibodies, Sarah Bray and Alexander Stark for plasmids, Anisha Kubasik-Thayil from the IMPACT imaging facility, University of Edinburgh, for assistance with imaging and Luigi Zechini and members of Andrew Jarman's lab, for useful advice and reagents. The work was funded by a BBSRC research grant awarded to Barry Denholm (BB/N001281/1).

## Additional information

### Funding

| Funder | Grant reference number | Author |
| --- | --- | --- |
| Biotechnology and Biological Sciences Research Council | BB/N001281/1 | Barry Denholm |

The funders had no role in study design, data collection and interpretation, or the decision to submit the work for publication.

### Author contributions

Robin Beaven, Conceptualization, Funding acquisition, Validation, Investigation, Visualization, Project administration, Writing—review and editing; Barry Denholm, Conceptualization, Investigation, Writing—original draft

### Author ORCIDs

Robin Beaven (iD) http://orcid.org/0000-0001-5941-1187
Barry Denholm (iD) http://orcid.org/0000-0001-5785-2552

### Decision letter and Author response

Decision letter https://doi.org/10.7554/eLife.35373.017
Author response https://doi.org/10.7554/eLife.35373.018

## Additional files

### Supplementary files

• Transparent reporting form
DOI: https://doi.org/10.7554/eLife.35373.015

### Data availability

The authors confirm that the data supporting the findings of this study are available within the article and its supplementary material.

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
