## [Decision Letter]

[Editors’ note: this article was originally rejected after discussions between the reviewers, but the authors were invited to resubmit after an appeal against the decision.]

Thank you for submitting your work entitled "Wingless acts as a long-range morphogen to pattern the proximo-distal axis of *Drosophila* renal tubules" for consideration by *eLife*. Your article has been reviewed by three peer reviewers, and the evaluation has been overseen by a Reviewing Editor and a Senior Editor. The reviewers have opted to remain anonymous.

Our decision has been reached after consultation between the reviewers. Based on these discussions and the individual reviews below, we regret to inform you that your work will not be considered further for publication in *eLife*.

These are nice data and in agreeing to have it reviewed we hoped that its advances and contributions will stand out. However, in our discussions following review, we were not compelled in this direction. Although Alexandre et al. showed that Wg can be functional when tethered to the membrane, it was never questioned that a gradient of a secreted Wg can be important. We expected that the developmental contexts may reveal more nuanced aspects of Wg's role as a gradient on in the tubules. We hope our reviews will be helpful in submitting to a specialised journal.

Reviewer #1:

I think this is an interesting paper that addresses an important issue with broad implications. Basically, the authors assess that a tethered Wingless cannot fully rescue the lack of wild-type Wingless in Malpighian tubule patterning; from this result the authors conclude that Wingless acts as a long-range morphogen in this process. Precisely because of the important implications of their statement I think it is important to dismiss alternative interpretations of the results before backing a specific conclusion.

In this regard, a first issue that should be convincingly addressed is whether the same amount of Wingless is produced both in the wild-type and by the tethered Wingless construct, as a quantitative difference rather than a qualitative difference might also account for the results reported. The authors may employ the tools already used to determine the endogenous Wingless to examine the tethered Wingless.

Besides, the same notion of long-range is at stake in this case as the authors indicate that the distal region of the tubule bents to approach the source of Wingless signal in the midgut. In this scenario, a cell-cell contact mediating the midgut-to-tubule wingless signalling can be discarded? Such a cell-to-cell signalling might also be modified by an alteration in Wingless subcellular localisation. The same above-mentioned tools would be also useful to determine the localisation of Wingless upon expression of the tethered form.

On a different note, the authors should provide higher magnifications of the images of the proximal tubule. The changes in expression patterns reported are very difficult to visualise in the present figures.

Reviewer #2:

In this manuscript, the authors revisit a problem that is under discussion since the publication of a paper in 2014 (Alexandre et al.), in which it was shown that a membrane-tethered Wingless protein can replace secreted Wingless to pattern the wing imaginal discs and gives rise to viable flies. This paper challenged the concept of Wingless being a morphogen, a molecule that activates target gene expression in a concentration dependent manner. Here, the authors present results, from which they conclude that Wingless acts as a morphogen to pattern the proximo-distal axis of the Malpighian tubules during *Drosophila* embryogenesis.

Their major findings are:

1) Inhibition of Wingless activity by expression of a dominant-negative form of Pangolin/dTCF using ct-Gal4 supresses expression of the target gene Odd in proximal tubules. In contrast, over-activation of the pathway extends the expression domain of Odd towards more distal, although not throughout the length of the tubule. As revealed by the analysis of Caudal expression, Wg is not the only signal responsible for proximo-distal patterning, since Caudal expression is not affected upon changes in Wg activity.

2) They confirmed that Wg is expressed in a small area of the midgut, close to the insertion site of the MT, and partially overlaps with Odd expression, but Odd expression extends a little further into the tubule.

3) They further show that activation of an Odd reporter gene by Wg depends on an enhancer element of Odd, which has recently been shown to be required for Odd activation by pan/dTCF in Kc cells. Expression of this reporter, which mimics endogenous Odd expression, is abolished upon mutation of the pan/dTCF binding site in this enhancer.

4) Finally, the authors show that membrane-tethered Wg, expressed in a wg mutant embryo, is unable to restore the expression pattern of Odd and results in abnormal ureter morphology.

Taken together, the authors confirm that at least in the MT, a graded expression of Wg seems to be required for normal ureter development.

From my point of view, some questions are not addressed, or some experiments are missing to make their conclusion solid.

a) The authors argue that *Nrt-wg* "is unable to correctly pattern the proximal tubule". This is shown by a reduced expression domain of Odd. It is not clear what "control" embryos are. Are these wild-type embryos or wild-type embryos expressing *Nrt-wg*? If these are just wild-type embryos, how can the authors exclude any dominant effect induced by *Nrt-wg*?

b) The same concern I have with the results presented in Figure 4. Is the control just a wild-type? Here it would also be interesting to know the phenotype obtained in the absence of Wg. Similarly: does secreted Wg rescue the loss-of-function phenotype? And does the phenotype shown represent a "patterning phenotype"? For me, it is more severe, since there also seems to be a non-autonomous effect on the visceral mesoderm, and ureter shortening could be the result of apoptosis.

c) From results presented in Figure 3 and 4 the authors conclude that Wg acts as long-range morphogen to pattern the tubules. One feature of a morphogen is its ability to activate target genes in a concentration dependent manner (as shown for Wg in the wing imaginal disc). Showing an effect on the expression of one target gene is, from my point of view, not sufficient to talk about "a long-range morphogen". In addition, the phenotype presented looks more severe than a mere patterning defect.

Reviewer #3:

Short but interesting paper reporting that in the Malpighian tubules, Wingless is secreted and disperses, as it does in most other developmental contexts. In the Malpighian tubules, however, this dispersal appears to be functionally important, demonstrating that different developmental contexts have different requirements for dispersal. I do not have any major concerns.

---

## [Author Response]

[Editors’ note: the author rebuttal and response to reviewers follow.]

Thank you for your time in assessing our manuscript "Wingless acts as a long-range morphogen to pattern the proximo-distal axis of *Drosophila* renal tubules," and for your efforts in thoroughly considering its potential suitability for eLife publication. We do however disagree with the assessment of the significance of this manuscript, and in particular the basis on which your decision not to publish it was made. You state that “it was never questioned that a gradient of a secreted Wg can be important” in the Alexandre et al study, but we would argue that this is precisely what they called into question. Whilst Alexandre et al noted that released Wingless is important for some aspects of fly development, they found no evidence for this in the context of tissue patterning – in which Wingless and other Wnts have been considered important – but rather that released Wingless plays a role in cell proliferation. This distinction is clearly not trivial and is reflected by publication of this study in Nature and its large number of citations. As they state, “we suggest therefore that the spread of Wingless is dispensable for patterning and growth even though it probably contributes to increasing cell proliferation”. They go on to argue that “the requirement for long-range spreading of other Wnts should be revisited”. Their work has therefore been cited in the field as a challenge to the assumption that Wingless and Wnts behave as released long-range morphogens, a question that is still contested. To give a few examples: “This [the Alexandre et al study] challenges the requirement for a spatial gradient of Wg” (Briscoe and Small, 2015). “Wingless… protein forms a long-range gradient in dorsoventral patterning of the wing disc of Drosophila. However, it should be carefully considered whether Wnt proteins act as long-range morphogens in vertebrates as well as Drosophila [reference to Alexandre et al paper]” (Mii, et al., 2017). “One of the reasons why it is important to take stock of this, is because there are few experiments (any?) in vertebrates that show a requirement for long range diffusion of Wnt proteins in pattern formation” (from “In time of revision: of Wingless and morphogens” - discussion article in the Node by Alfonso Martinez Arias). Indeed, as reviewer #2 asserts “this paper challenged the concept of Wingless being a morphogen, a molecule that activates target gene expression in a concentration dependent manner”, and the view that this question is still contested is also supported by their comment that in our manuscript we “revisit a problem that is under discussion since the publication of a paper in 2014 (Alexandre et al.)”.

In light of this we would argue that our findings are both important and of high interest to the Wnt/Wingless field. The Alexandre et al study provides a good example of how important it is for scientific advance to be made on the basis of empirical evidence, rather than upon assumptions, correlative evidence etc. As far as we are aware our study is the first demonstration that experimentally restricting the release of a Wingless or Wnt member results in defective patterning and morphogenesis, therefore making it an important advance. The continuing contested nature of this issue will make it of high interest to the field. Furthermore, the Alexandre et al study has also informed broader discussions about the nature of morphogens and the different modes by which signals can shape development (Briscoe and Small, 2015; Bryant and Gardiner, 2016; Sagner and Briscoe, 2017), and our findings will therefore also be of broader interest, informing such discussions. In support of this, reviewer #3 describes our work as a “short but interesting paper” and reviewer #1 states “I think this is an interesting paper that addresses an important issue with broad implications”.

Overall, the reviewers’ comments support our view that the role secreted Wingless plays in pattering remains a debated issue, and that in light of this our study is important and interesting. Reviewer #3 had no major concerns, whilst we feel that the useful concerns and suggestions of the other reviewers can be addressed which would significantly strengthen our manuscript. We would be extremely grateful if you could reconsider your decision in light of the points raised above. Thank you again for your efforts in critically evaluating our manuscript.

Reviewer #1:I think this is an interesting paper that addresses an important issue with broad implications. Basically, the authors assess that a tethered Wingless cannot fully rescue the lack of wild-type Wingless in Malpighian tubule patterning; from this result the authors conclude that Wingless acts as a long-range morphogen in this process. Precisely because of the important implications of their statement I think it is important to dismiss alternative interpretations of the results before backing a specific conclusion.

We are thankful for reviewer #1’s positive view of our study, regarding the importance of the issue it addresses, as well as its interest and broader implications. We agree with the need to dismiss alternative interpretations, and believe we have been able to do so, incorporating the useful suggestions made by reviewer #1 to these ends.

In this regard, a first issue that should be convincingly addressed is whether the same amount of Wingless is produced both in the wild-type and by the tethered Wingless construct, as a quantitative difference rather than a qualitative difference might also account for the results reported. The authors may employ the tools already used to determine the endogenous Wingless to examine the tethered Wingless.

We agree that it is important to assess possible effects of the addition of the Nrt fusion protein upon the levels of Wg, to address this concern. We have now stained *Nrt-wg* and wild-type control embryos with anti-Wingless and incorporated this into the manuscript (Figure 4—figure supplement 1). We observe a stronger anti-Wingless staining in the posterior midgut of *Nrt-wg* embryos. This is likely to result at least in part from the lack of diffusion of the Nrt-Wg protein, resulting in its higher concentration in its originating cells. We consider this finding to provide strong evidence that the *Nrt-wg* phenotypes we observe cannot be explained by a reduced presence of Wg protein in this context. It is conceivable that excessive levels of Nrt-Wg protein lead to some form of dominant disruptive effect, however other lines of evidence argue against this as we discuss in our response to reviewer #2’s comments below.

Besides, the same notion of long-range is at stake in this case as the authors indicate that the distal region of the tubule bents to approach the source of Wingless signal in the midgut. In this scenario, a cell-cell contact mediating the midgut-to-tubule wingless signalling can be discarded? Such a cell-to-cell signalling might also be modified by an alteration in Wingless subcellular localisation. The same above-mentioned tools would be also useful to determine the localisation of Wingless upon expression of the tethered form.

We feel that it is likely that cell-cell signalling is not disrupted in *Nrt-wg* flies as Wingless targets have been shown to be activated in cells adjoining Nrt-Wg expressing cells in the other contexts of the fly wing imaginal discs (Alexandre et al., 2014; Zecca and Struhl, 2010; Zecca et al., 1996) and in bristle differentiation (Zecca et al., 1996), as we have now pointed out in the manuscript. Staining *Nrt-wg* embryos with anti-Wg as suggested revealed Nrt-Wg protein was localised to the plasma membrane as expected (Figure 4—figure supplement 1). Nrt-Wg appears relatively evenly around each cell, which suggests it is in the correct location to induce Wg signalling in adjoining cells. It does also appear from these images that the domain of Odd skipped expressing cells is larger than the domain of cells expressing Nrt-Wg, which also supports the view that Nrt-wg can induce signalling in neighbouring cells in this tissue.

On a different note, the authors should provide higher magnifications of the images of the proximal tubule. The changes in expression patterns reported are very difficult to visualise in the present figures.

Original Figure 1 has now been divided in two to give more space, with this suggestion in mind. Enlarged regions have now been added to figures to address this point.

Reviewer #2:In this manuscript, the authors revisit a problem that is under discussion since the publication of a paper in 2014 (Alexandre et al.), in which it was shown that a membrane-tethered Wingless protein can replace secreted Wingless to pattern the wing imaginal discs and gives rise to viable flies. This paper challenged the concept of Wingless being a morphogen, a molecule that activates target gene expression in a concentration dependent manner. Here, the authors present results, from which they conclude that Wingless acts as a morphogen to pattern the proximo-distal axis of the Malpighian tubules during Drosophila embryogenesis.Their major findings are:1) Inhibition of Wingless activity by expression of a dominant-negative form of Pangolin/dTCF using ct-Gal4 supresses expression of the target gene Odd in proximal tubules. In contrast, over-activation of the pathway extends the expression domain of Odd towards more distal, although not throughout the length of the tubule. As revealed by the analysis of Caudal expression, Wg is not the only signal responsible for proximo-distal patterning, since Caudal expression is not affected upon changes in Wg activity.2) They confirmed that Wg is expressed in a small area of the midgut, close to the insertion site of the MT, and partially overlaps with Odd expression, but Odd expression extends a little further into the tubule.3) They further show that activation of an Odd reporter gene by Wg depends on an enhancer element of Odd, which has recently been shown to be required for Odd activation by pan/dTCF in Kc cells. Expression of this reporter, which mimics endogenous Odd expression, is abolished upon mutation of the pan/dTCF binding site in this enhancer.4) Finally, the authors show that membrane-tethered Wg, expressed in a wg mutant embryo, is unable to restore the expression pattern of Odd and results in abnormal ureter morphology.Taken together, the authors confirm that at least in the MT, a graded expression of Wg seems to be required for normal ureter development.From my point of view, some questions are not addressed, or some experiments are missing to make their conclusion solid.

We share the view of reviewer #2 that the problem revisited by our study is one that has remained under discussion since the publication of the Alexandre et al. study in 2014, and we therefore anticipate that our findings will be of significant interest in informing this ongoing discussion. We find the suggested experiments highly useful in strengthening the conclusions of our study. We consider that we have been able to address these concerns, modifying our manuscript accordingly as outlined below.

a) The authors argue that Nrt-wg "is unable to correctly pattern the proximal tubule". This is shown by a reduced expression domain of Odd. It is not clear what "control" embryos are. Are these wild-type embryos or wild-type embryos expressing Nrt-wg? If these are just wild-type embryos, how can the authors exclude any dominant effect induced by Nrt-wg?

The control genotypes have now been clearly stated in the figure legends. In this case (Figure 4 of the revised manuscript) control embryos are ‘sibling controls’ (the *Nrt-wg* stock is kept over a YFP balancer and YFP expressing embryos were used for control, i.e. a mixture of *Nrt-wg*/balancer YFP and balancer YFP/balancer YFP embryos). It is conceivable that Nrt-Wg could have a dominant effect, although it should be noted that Nrt-Wg functions similarly to wild-type Wg in the contexts of larval denticle patterning, wing, leg and eye imaginal disc patterning, and bristle differentiation (Alexandre et al., 2014; Zecca et al., 1996). Early functions of Wg in Malpighian tubule specification and cell proliferation, and activation of Odd skipped in the posterior midgut. However, to test whether a dominant effect could be induced by *Nrt-wg* expression we have carried out an additional set of control experiments in which we used *wg-Gal4* to drive the expression of *UAS-Nrt-wg* in a background where the endogenous *wg* is present (Figure 4—figure supplement 2). This does not lead to a reduced Odd-expression domain in the tubule, and we consider this to be strong evidence that the *Nrt-wg* phenotypes are not a dominant effect of the Nrt-Wg protein.

b) The same concern I have with the results presented in Figure 4. Is the control just a wild-type?

In this case heterozygous *Nrt-wg* adults were used for the control; the tubules/ureters are indistinguishable from wild-type. This provides additional evidence that the defects we observe do not result from a dominant effect of the presence of Nrt-Wg, as at least the presence of one copy does not induce a phenotype. Unfortunately we were unable to also get data from *wg-Gal4>UAS-Nrt-wg*. These flies do not survive until adulthood, which is likely to be due to developmental defects resulting from a delayed temporal control of expression for these tools (e.g. extended Wg activity due to perdurance from UAS-GAL4 tools).

Here it would also be interesting to know the phenotype obtained in the absence of Wg.

Homozygous *wg^1-12^* flies (exposed to the temperature shift regimes described in the ms) did not reach adulthood due to a role for wingless in post-embryonic development. Neither did *UAS-pan-∆N/ CtB-Gal4;CtB-Gal4/+* which appear to die as early larvae, possibly due to aberrant renal function. However *UAS-arm^S10^/ CtB-Gal4;CtB-Gal4/+* (i.e. driving constitutive activity of Wg signalling) flies did survive to adulthood and we observed that the ensheathing visceral muscle extends more distally along the tubule (Figure 5—figure supplement 1). This phenotype is in the opposite direction to that observed in *Nrt-wg* adults, and supports the hypothesis (see below) that changing the identity of the tubule cells has a non-autonomous effect upon their recognition by visceral mesoderm during development.

Similarly: does secreted Wg rescue the loss-of-function phenotype?

The ideal experiment here would be to test rescue of *Nrt-wg/Nrt-wg* animals with *wg-Gal4>UAS-wg*, however this is not technically possible as *wg-Gal4* is inserted into the *wg* locus (as is *Nrt-wg*). However, as discussed above we consider it unlikely that *Nrt-wg* has a dominant effect (response to point a, above).

And does the phenotype shown represent a "patterning phenotype"? For me, it is more severe, since there also seems to be a non-autonomous effect on the visceral mesoderm, and ureter shortening could be the result of apoptosis.

These are interesting points regarding the interpretation of the *Nrt-wg* phenotype upon the morphology of the adult tubules. We consider that the effect upon the visceral muscle which ensheaths the ureter to be a non-autonomous effect. To our view it could be explained by the identity of the cells of the more distal section of ureter being altered as a result of not expressing Odd skipped (and potentially other genes) and consequently not being recognised by migrating visceral mesoderm during development. We therefore consider this the secondary result of a patterning defect. We have now made these ideas explicit in the text of the manuscript. The idea that the phenotype could result from apoptosis is an interesting one. In order to directly test whether apoptosis is contributing to the observed phenotype, we stained *Nrt-wg* embryos with an antibody marker of apoptotic cells. We did not observe any clear cases of apoptotic cells in the proximal tubule indicating that cell death is not elevated in the tubule in this genetic context (Figure 4—figure supplement 3). As the ureters are much wider (as well as shorter) we suggests the phenotype is more likely a result of distorted morphology rather than a simple loss of cells. Further, our interpretation as outlined above is that the distal section of ureter is present, but is altered in identity – in that it lacks the ensheathing visceral muscle.

c) From results presented in Figure 3 and 4 the authors conclude that Wg acts as long-range morphogen to pattern the tubules. One feature of a morphogen is its ability to activate target genes in a concentration dependent manner (as shown for Wg in the wing imaginal disc). Showing an effect on the expression of one target gene is, from my point of view, not sufficient to talk about "a long-range morphogen".

We have altered our wording in the title and throughout the manuscript to stress our essential conclusion, that the patterning role of Wg does require release and spread in the context of the Malpighian tubule. We consider our findings to point towards Wg acting as a morphogen in this context, however we agree with reviewer #2’s assessment that more than one target gene patterning several distinct regions would be required to draw this conclusion. Future work, for example identifying further Wg target genes in the tubule, will be necessary to resolve this issue.

In addition, the phenotype presented looks more severe than a mere patterning defect.

We disagree with the idea that patterning defects cannot result in severe phenotypes (indeed many of the most striking *Drosophila* phenotype result from patterning defects e.g. Hox mutants). Our argument is that the ureter cells have an altered identity based on defects during developmental patterning. We do not know what this new identity is (more distal tubule?) but we might expect the change to have severe phenotypic consequences, including non-autonomous effects as outlined above.

Reviewer #3:Short but interesting paper reporting that in the Malpighian tubules, Wingless is secreted and disperses, as it does in most other developmental contexts. In the Malpighian tubules, however, this dispersal appears to be functionally important, demonstrating that different developmental contexts have different requirements for dispersal. I do not have any major concerns.

We are pleased to hear that reviewer #3 evaluates this to be an interesting paper and has no major concerns with it.